# Gill-associated ammonia oxidizers are widespread in teleost fish

Wouter Mes,[1,2] Sebastian Lücker,[1] Mike S.M. Jetten,[1] Henk Siepel,[2] Marnix Gorissen,[2] Maartje A.H.J. van Kessel[1]

**ABSTRACT** Recent advances in sequencing methods have greatly expanded the knowledge of teleost-associated microorganisms. While fish-gut microbiomes are comparatively well studied, less attention has gone toward other, external organ-microbiome associations. Gills are particularly interesting to investigate due to their functions in gas exchange, osmoregulation, and nitrogen excretion. We recently discovered a branchial symbiosis between nitrogen-cycling bacteria and teleosts (zebrafish and carp), in which ammonia-oxidizing *Nitrosomonas* and denitrifying bacteria together convert toxic ammonia excreted by the fish into harmless dinitrogen ($N_2$) gas. This symbiosis can function as a "natural biofilter" in fish gills and can potentially occur in all ammonotelic fish species, but it remains unknown how widespread this symbiosis is. In this study, we analyzed all publicly available gill microbiome data sets and checked for the presence of *Nitrosomonas*. We discovered that more than half of the described fish gill microbiomes contain 16S rRNA gene sequences of ammonia-oxidizing bacteria (AOB). The presence of gill-specific AOB was shown in both wild and aquacultured fish, as well as in marine and freshwater fish species. Based on these findings, we propose that ammonia oxidizers are widespread in teleost fish gills. These gill-associated AOB can significantly affect fish nitrogen excretion, and the widespread nature of this association suggests that the gill-associated AOB can have similar impacts on more fish species. Future research should address the contribution of these microorganisms to fish nitrogen metabolism and the fundamental characteristics of this novel symbiosis.

**IMPORTANCE** Recent advances in sequencing have increased our knowledge of teleost-associated microbiota, but the gill microbiome has received comparatively little attention. We recently discovered a consortium of nitrogen-cycling bacteria in the gills of common carp and zebrafish, which are able to convert (toxic) ammonia into harmless dinitrogen gas. These microorganisms thus function as a natural nitrogen biofilter. We analyzed all available gill microbiome data sets to determine how widespread gill-associated ammonia-oxidizing bacteria (AOB) are. More than half of the data sets contained AOB, representing both aquacultured and wild fish from freshwater and marine habitats. In total, 182 amplicon sequencing variants were obtained, of which 115 were found specifically in the gills and not the environmental microbiomes. As gill-associated AOB are apparently widespread in teleost fish, it is important to study their impact on host nitrogen excretion and the potential to reduce ammonia accumulation in (recirculating) aquaculture of relevant fish species.

**KEYWORDS** gill, microbiome, symbiont

Address correspondence to Wouter Mes, wouter.mes@ru.nl, or Maartje A.H.J. van Kessel, maartje.vankessel@ru.nl.

The authors declare no conflict of interest.

See the funding table on p. 5.

Animals evolved in a bacterial world and are profoundly shaped by their microbiomes (1). Teleost fish, representing more than half of all vertebrate species, are an interesting target for exploring host-microbiome dynamics and recent developments in next-generation sequencing have expanded our knowledge of the fish microbiome, particularly in the gut (2, 3). By contrast, little attention has gone to other organs, even though these are likely to be equally important for organismal health and physiology

(4, 5). Moreover, a *functional* understanding of fish-associated microorganisms is in its infancy (6).

An example of a functional exploration of specific bacteria living in association with fish is the recently discovered symbiosis of nitrogen-cycling bacteria in the gills of carp and zebrafish (7). A consortium consisting of an ammonia-oxidizing *Nitrosomonas* and an as-of-yet unidentified denitrifying microorganism converts ammonia produced by the fish into dinitrogen gas ($N_2$), thereby functioning as a natural, nitrogen-removing biofilter. Although activity measurements have so far only been performed in the aforementioned species, the presence of *Nitrosomonas* was shown by amplicon sequencing in other fish species, including Atlantic salmon (*Salmo salar*), yellowtail kingfish (*Seriola lalandi*), and red snapper (*Lutjanus campechanus*) (8–11), although none of these studies investigated the potential of these bacteria to remove ammonia *in vivo*. To explore how widespread this symbiosis potentially is in fish, we analyzed all publicly available gill microbiome data sets published for the presence of AOB.

We analyzed 44 data sets that included gill samples, coming from a wide variety of fish species and environments (Table S1). Data were analyzed using the DADA2 pipeline to obtain amplicon sequencing variants (ASVs), and taxonomy was assigned to ASVs using the SILVA database (version 138) (12, 13). We recorded the total number of ammonia oxidizer ASVs in each data set (including samples from other fish organs and environmental samples), as well as how many of these were present in gill samples, to obtain an overview of gill-associated AOB. Sequences were placed on a maximum-likelihood tree containing aquacultured AOB species full-length 16S rRNA genes using *pplacer* to observe taxonomic relationships between gill-associated ASVs (14). This allows a comparison of ASVs from different variable regions of the 16S rRNA gene.

Of the analyzed data sets, 59% contained ASVs of the ammonia oxidizers *Nitrosomonas* or *Nitrosospira* (Table S2), while no gammaproteobacterial AOB were found. Gill-associated AOB were found in fish from freshwater and marine habitats (11 and 16 species, respectively) and in both wild and farmed fish (14 and 12 species, respectively). In total, 182 ASVs were obtained from the combined data sets, which were placed in the full-length 16S rRNA gene tree of Nitrosomonadaceae and closely related taxa (Fig. 1). The presence of ammonia oxidizers in this wide range of fish species from diverse environments shows that the occurrence of gill-associated AOB is widespread in multiple habitats.

Since the data sets were obtained through different sampling methods, DNA extraction protocols, primer combinations, and sequencing platforms (Table S1), it is not possible to directly compare data from different studies. Primer combinations targeting different variable regions of the 16S rRNA gene do not seem to affect the detection of gill-associated AOB. No trends are apparent between wild and aquacultured fish regarding the AOB taxonomy and the type of aquaculture system (recirculating aquaculture systems [RAS] vs. flow-through) did not notably influence the presence of gill-associated AOB. However, based on the taxonomy of the ammonia oxidizers observed in freshwater and marine fish (Fig. 1), it is clear that some lineages of *Nitrosomonas* are exclusive to freshwater or marine habitats. No ASVs derived from marine fish species were placed on the *Nitrosomonas europaea*, *Nitrosomonas eutropha*, and *Nitrosomonas communis* branches of the tree, while freshwater fish species had 58 ASVs clustering with these lineages. Thus, salinity seems to affect the AOB species associated with the gills. Salt tolerance varies considerably between *Nitrosomonas* species and is one of the key environmental parameters influencing which species are present (15). For horizontal bacterial transmission, (temporary) survival outside the fish gill is required, indicating that gill-associated nitrifiers are adapted to the salinity of the water.

Due to the potential need for survival of gill-associated AOB in the water, it is interesting to investigate the relationship between environmental and gill-associated ammonia oxidizers. Whenever environmental samples were available, the presence of AOB in the surrounding water was checked (Table S2). In several studies, there was considerable overlap between the environmental and gill AOB community. In particular,

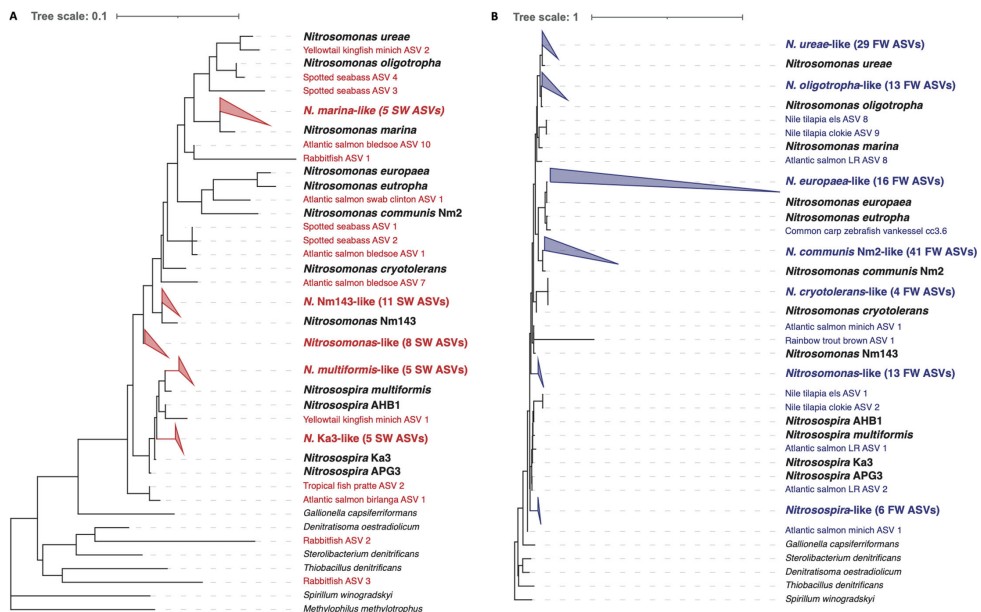

**FIG 1** Phylogenetic placement of gill-associated nitrifier amplicon sequencing variants (ASVs). (A) Phylogenetic placement of nitrifier ASVs (*n* = 49) from fish in marine habitats on a maximum-likelihood full-length 16S rRNA gene reference tree. ASVs clustering with a specific nitrifier lineage are collapsed into a cluster with the name of the nitrifier species. (B) Phylogenetic placement of nitrifier ASVs (*n* = 133) from fish in freshwater habitats on a maximum-likelihood full-length 16S rRNA gene reference tree. ASVs clustering with a specific nitrifier lineage are collapsed into a cluster with the name of the nitrifier species. Phylogenetic placement was performed with the *pplacer* algorithm, and tree visualization was performed with iTOL.

there seem to be few gill-*specific* AOB in RAS (16). This is of interest, considering the reliance of RAS on active ammonia-oxidizing microorganisms in the biofilter (10). However, in other data sets, gill-associated nitrifiers were identified while none were found in the surrounding water (17, 18), which suggests selective colonization by gill-*specific* AOB. In all, we identified 115 AOB ASVs that were present in gill samples but not in the environment (water, sediment, or rearing system), constituting the majority of ASVs found (Fig. 2; Table S2). While the lack of AOB observed in environmental samples is not necessarily evidence of their complete absence, this observation suggests that specific ammonia oxidizers are colonizing the gills.

While several studies have noticed gill-associated AOB in the past, most studies included here do not mention their presence in the gill. A reason for this could be the low relative abundance of gill-associated nitrifiers. Although the relative abundance reached more than 50% in some gill samples of Atlantic salmon (*S. salar* L.) (11), the relative abundance in the investigated data sets is low (<1%). This does not necessarily translate to a small impact as it was shown that nitrogen-cycling microorganisms impact the nitrogen balance of fish significantly (19), regardless of the apparent low relative abundance (20). It is known that sampling methods can reduce the apparent abundance of microorganisms in tissues with low bacterial biomass (17). In order to assess the true functional impact of gill-associated AOB on the fish, it will be key to relate AOB abundance to measurable ammonia-oxidizing activity. Additional approaches are required to link the abundance of nitrifying bacteria to their activity and establish the impact on fish physiology.

Considering the widespread presence of gill-associated AOB in teleost fish species, it is important to establish the effects of these ammonia oxidizers on host health and nitrogen physiology. Additionally, it is worthwhile to study the potential of the gill-associated AOB in reducing ammonia accumulation, which is a common issue in aquaculture (21). From a fundamental perspective, studying this functional interaction between nitrogen-cycling microorganisms and a vertebrate host is also highly interesting. These

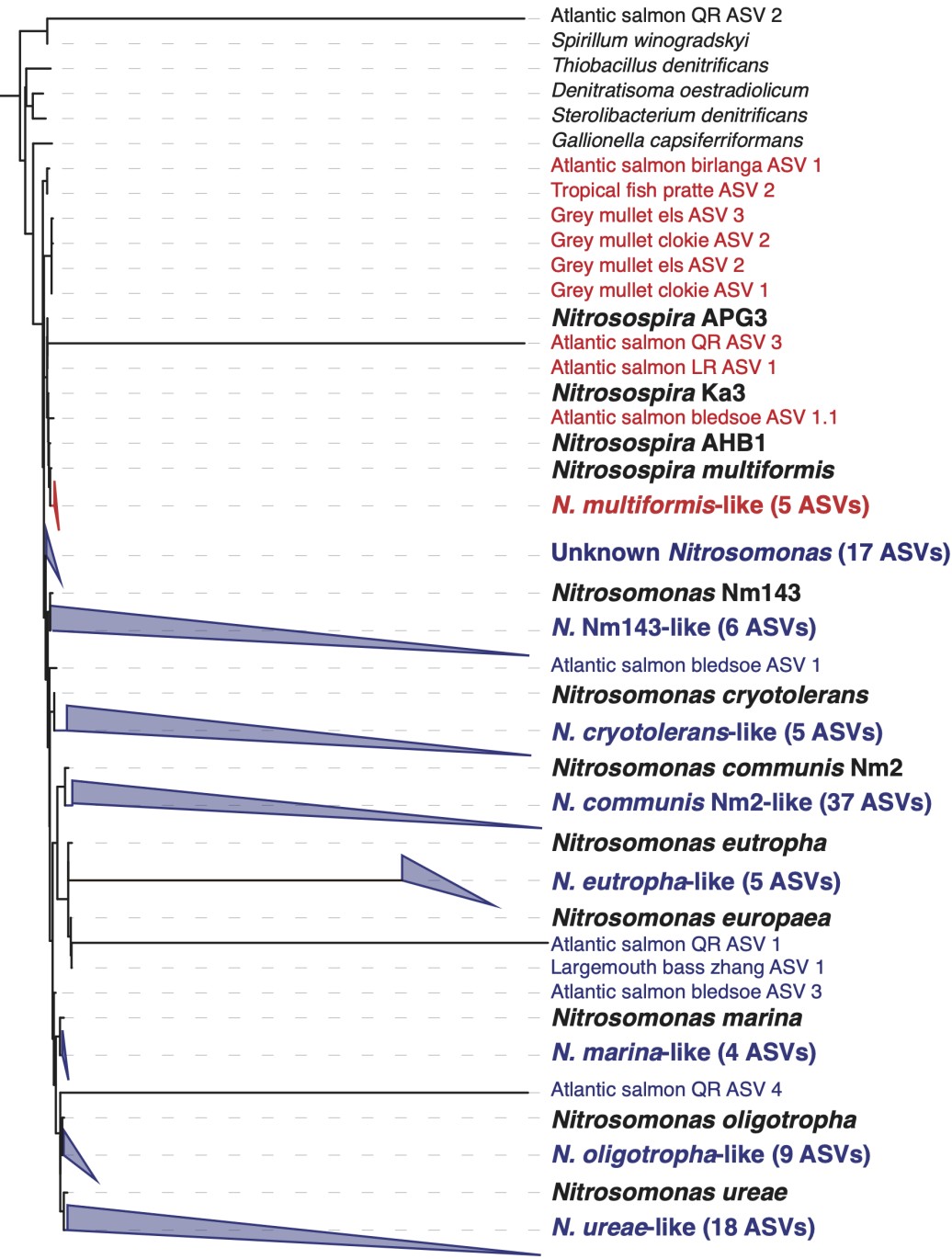

**FIG 2** Phylogenetic placement of gill-specific nitrifier amplicon sequencing variants (ASVs). ASVs of gill-associated nitrifiers were compared with environmental nitrifiers to obtain ASVs that were only found in gill samples and not environmental samples. The remaining gill-specific ASVs ($n = 115$) were placed on a full-length 16S rRNA gene maximum-likelihood tree. ASVs clustering in the *Nitrosomonas* genus are indicated in blue, while ASVs clustering in the *Nitrosospira* genus are indicated in red. ASVs clustering with a specific nitrifier lineage are collapsed into a cluster with the name of the nitrifier species. Phylogenetic placement was performed with the *pplacer* algorithm, and tree visualization was performed with iTOL.

gill-associated AOB can significantly affect fish nitrogen excretion, and the widespread nature of this association suggests that they can have similar impacts on more fish species. New studies should investigate the link between the abundance and activity of the nitrogen-cycling bacteria in these fish species.

## AUTHOR AFFILIATIONS

[1]Department of Microbiology, Radboud Institute for Biological and Environmental Sciences, Radboud University, Nijmegen, the Netherlands
[2]Department of Plant & Animal Biology, Radboud Institute for Biological and Environmental Sciences, Radboud University, Nijmegen, the Netherlands

## AUTHOR ORCIDs

Wouter Mes http://orcid.org/0000-0001-7815-6687
Mike S.M. Jetten http://orcid.org/0000-0002-4691-7039
Maartje A.H.J. van Kessel http://orcid.org/0000-0003-0271-6478

## FUNDING

| Funder | Grant(s) | Author(s) |
|---|---|---|
| Nederlandse Organisatie voor Wetenschappelijk Onderzoek (NWO) | Spinoza premie 2012 | Mike S. M. Jetten |
| Nederlandse Organisatie voor Wetenschappelijk Onderzoek (NWO) | Gravitation grant SIAM 024.002.002 | Mike S. M. Jetten |
| Nederlandse Organisatie voor Wetenschappelijk Onderzoek (NWO) | 016.Vidi.189.050 | Sebastian Lücker |
| Nederlandse Organisatie voor Wetenschappelijk Onderzoek (NWO) | 016.Veni.192.062 | Maartje A.H.J. van Kessel |
| EC | Erasmus+ (Еразъм+) | ACES+ | Marnix Gorissen |

## AUTHOR CONTRIBUTIONS

Wouter Mes, Conceptualization, Formal analysis, Investigation, Writing – original draft | Sebastian Lücker, Methodology, Supervision, Writing – review and editing | Mike S.M. Jetten, Supervision, Writing – review and editing | Henk Siepel, Supervision, Writing – review and editing | Marnix Gorissen, Funding acquisition, Investigation, Supervision, Writing – review and editing | Maartje A.H.J. van Kessel, Conceptualization, Investigation, Supervision, Writing – review and editing

## ADDITIONAL FILES

The following material is available online.

### Supplemental Material

**Supplemental material (Spectrum00295-24-s0001.docx).** Tables S1 and S2.

### Open Peer Review

**PEER REVIEW HISTORY (review-history.pdf).** An accounting of the reviewer comments and feedback.

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
