## [Reviewer comments · Microbiology Spectrum]

Microbiology Spectrum

Gill-associated ammonia oxidizers are widespread in teleost fish

Wouter Mes, Sebastian Lücker, Mike Jetten, Henk Siepel, Marnix Gorissen, and Maartje van Kessel

Corresponding Author(s): Wouter Mes, Radboud Universiteit

Review Timeline:

Submission Date:	February 2, 2024
Editorial Decision:	July 8, 2024
Revision Received:	July 29, 2024
Accepted:	August 28, 2024

Editor: Luke Iwanowicz

Reviewer(s): The reviewers have opted to remain anonymous.

Transaction Report:

DOI: <https://doi.org/10.1128/spectrum.00295-24>

Re: Spectrum00295-24 (Gill-associated ammonia oxidizers are widespread in teleost fish)

Dear Dr. Wouter Mes:

Thank you for the privilege of reviewing your work. Below you will find my comments, instructions from the Spectrum editorial office, and the reviewer comments.

We have now received two reviews for this manuscript. Both reviewers found the content interesting and of interest to the journal readership. In general, the reviewer comments and requests were minor.

Revision Guidelines

Sincerely,
Luke Iwanowicz
Editor
Microbiology Spectrum

Reviewer #1 (Comments for the Author):

The authors use published datasets to identify the presence of Nitrosomonas in the fish gills in order to delineate whether symbiosis between those groups and fish gills is widespread among the teleost group. Overall I think it is a nice approach, but more information is needed before drawing conclusions.

Line 65 and Supplementary Table S1: Which variable region did the authors choose for their analysis? Were all regions included? Please mention in the table which regions were targeted in the different studies and how this was taken into account during analysis. This is because the region can also have a certain bias in the results in terms of diversity and abundance. Also which of all these studies refer to wild or farmed fish? This is not clear, and I do think it may have a significant effect on the outcome you present, as well as whether the fish come from RAS or flow trough systems, where the dilution effect is even larger in terms of finding out Nitrosomonas in the surrounding environment. I am also missing information in the tree on whether there is a phylogenetic closeness between biofilter/water Nitrosomonas of the study for example and the gills in RAS. I am not sure if you have such studies in your dataset.

Reviewer #2 (Comments for the Author):

Thank you for the opportunity to review this observation of very intriguing findings regarding the presence of ammonia oxidizers associated with teleost fish gills. Overall, the manuscript was very well-written and the methodological approach appears to be valid, yielding findings of broad interest to the ASM community. I have made a few minor suggestions in the attached manuscript.

1 Gill-associated ammonia oxidizers are widespread in teleost fish

2 Wouter Mes^{1,2*}, Sebastian Lücker¹, Mike S.M. Jetten¹, Henk Siepel², Marnix Gorissen², Maartje A.H.J. van
3 Kessel^{1*}

4
5 ¹ Department of Microbiology, Radboud Institute for Biological and Environmental Sciences, Radboud
6 University, Nijmegen, the Netherlands

7 ² Department of Plant & Animal Biology, Radboud Institute for Biological and Environmental Sciences,
8 Radboud University, Nijmegen, the Netherlands

9 * Corresponding author

10

11 Abstract

12 Recent advances in sequencing methods have greatly expanded the knowledge of teleost-
13 associated microorganisms. While fish-gut microbiomes are comparatively well-studied, less
14 attention has gone towards other, external organ-microbiome associations. Gills are particularly
15 interesting to investigate, due to their functions in gas exchange, osmoregulation and nitrogen
16 excretion. We recently discovered a branchial symbiosis between nitrogen cycle bacteria and
17 teleosts (zebrafish and carp), in which ammonia-oxidizing *Nitrosomonas* and denitrifying bacteria
18 together convert toxic ammonia excreted by the fish into harmless dinitrogen (N₂) gas. This
19 symbiosis can function as a ‘natural biofilter’ in fish gills and can potentially occur in all
20 ammonotelic fish species, but it remains unknown how widespread this symbiosis is. In this
21 study, we analyzed all publicly available gill microbiome datasets and checked for the presence of
22 *Nitrosomonas*. We discovered that more than half of the described fish gill microbiomes contain
23 16S rRNA gene sequences of ammonia-oxidizing bacteria (AOB). The presence of gill-specific
24 AOB was shown in both wild and caged fish, as well as in marine and freshwater fish species.
25 Based on these findings, we propose that ammonia oxidizers are widespread in teleost fish gills.
26 These gill-associated AOB can significantly affect fish nitrogen excretion and the widespread
27 nature of this association suggests that the gill-associated AOB can have similar impacts in more
28 fish species. Future research should address the contribution of these microorganisms to fish
29 nitrogen metabolism and the fundamental characteristics of this novel symbiosis.

30

31 Importance

32 Recent advances in sequencing have increased our knowledge of teleost-associated microbiota,
33 but the gill microbiome has received comparatively little attention. We recently discovered a
34 consortium of nitrogen cycle bacteria in gills of common carp and zebrafish, which are able to
35 convert (toxic) ammonia into harmless dinitrogen gas. These microorganisms thus function as a

36 natural nitrogen biofilter. We analyzed all available gill microbiome datasets to determine how
37 widespread gill-associated ammonia-oxidizing bacteria (AOB) are. More than half of the datasets
38 contained AOB, representing both cultured and wild fish from freshwater and marine habitats.
39 In total, 182 amplicon sequencing variants (ASVs) were obtained, of which 115 were found
40 specifically in the gills and not the environmental microbiomes. As gill-associated AOB are
41 apparently widespread in teleost fish, it is important to study their impact on host nitrogen
42 excretion and the potential to reduce ammonia accumulation in (recirculating) aquaculture of
43 relevant fish species.

44

45 Animals evolved in a bacterial world and are profoundly shaped by their microbiomes (McFall-
46 Ngai et al. (2013). Teleost fish, representing more than half of all vertebrate species, are an
47 interesting target for exploring host-microbiome dynamics and recent developments in next-
48 generation sequencing have expanded our knowledge of the fish microbiome, particularly in the
49 gut (Ghanbari et al., 2015; Perry et al., 2020). By contrast, little attention has gone to other
50 organs, even though these are likely to be equally important for organismal health and physiology
51 (Pratte et al., 2018; Sehnal et al., 2021). Moreover, a *functional* understanding of fish-associated
52 microorganisms is in its infancy (Legrand et al., 2019).

53 An example of a functional exploration of specific bacteria living in association with fish
54 is the recently discovered symbiosis of nitrogen-fixing bacteria in the gills of carp and zebrafish
55 (van Kessel et al., 2016). A consortium consisting of an ammonia-oxidizing *Nitrosomonas* and an
56 as of yet unidentified denitrifying microorganism converts ammonia produced by the fish into
57 dinitrogen gas (N₂); thereby functioning as a natural, nitrogen-removing biofilter. Although
58 activity measurements have so far only been performed in the aforementioned species, the
59 presence of *Nitrosomonas* was shown by amplicon sequencing in other fish species, including
60 Atlantic salmon (*Salmo salar*), yellowtail kingfish (*Seriola lalandi*) and red snapper (*Lutjanus*
61 *campechanus*) (Legrand et al., 2018; Minich et al., 2020; Birlanga et al., 2022; Lorgen-Ritchie et al.,
62 2022) although none of these studies investigated the potential of these bacteria to remove
63 ammonia *in vivo*. To explore how widespread this symbiosis potentially is in fish, we analyzed all
64 publicly available gill microbiome datasets published for the presence of AOB.

65 We analyzed 44 datasets that included gill samples, coming from a wide variety of fish
66 species and environments (Table S1). Data were analyzed using the DADA2 pipeline to obtain
67 amplicon sequencing variants (ASVs), and taxonomy was assigned to ASVs using the SILVA
68 database (version 138) (Quast et al., 2012; Callahan et al., 2016). We recorded the total number
69 of ammonia oxidizer ASVs in each dataset (including samples from other fish organs and
70 environmental samples), as well as how many of these were present in gill samples, to obtain an
71 overview of gill-associated AOB. Sequences were placed on a maximum-likelihood tree
72 containing cultured AOB species 16S rRNA genes using *pplacer* to observe taxonomic
73 relationships between gill-associated ASVs (Matsen et al., 2010).

74 Of the analyzed datasets, 59% contained ASVs of the ammonia oxidizers *Nitrosomonas* or
75 *Nitrospira* (Table S2), while no gammaproteobacterial AOB were found. Gill-associated AOB
76 were found in fish from freshwater and marine habitats (11 and 16 species, respectively) and in
77 both wild and farmed fish (14 and 12 species, respectively). In total, 182 ASVs were obtained
78 from the combined datasets, which were placed in the full-length 16S rRNA gene tree of

79 Nitrosomonadaceae and closely related taxa (Fig. 1). The presence of ammonia oxidizers in this
80 wide range of fish species from diverse environments shows that the occurrence of gill-
81 associated AOB is widespread in multiple habitats.

82 Since the datasets were obtained through different sampling methods, DNA extraction
83 protocols, primer combinations and sequencing platforms (Table S1), it is not possible to directly
84 compare data from different studies. However, based on the taxonomy of the ammonia oxidizers
85 observed in freshwater and marine fish (Fig. 1), it is clear that some lineages of *Nitrosomonas* are
86 exclusive to freshwater or marine habitats. No ASVs derived from marine fish species were
87 placed on the *N. europaea*, *N. eutropha* and *N. communis* branches of the tree, while freshwater fish
88 species had 58 ASVs clustering with these lineages. Thus, salinity seems to affect the AOB
89 species associated with the gills. Salt tolerance varies considerably between *Nitrosomonas* species
90 and is one of the key environmental parameters influencing which species are present (Norton,
91 2011). For horizontal bacterial transmission, (temporary) survival outside the fish gill is required,
92 indicating that gill-associated nitrifiers are adapted to the salinity of the water.

93 Due to the potential need for survival of gill-associated AOB in the water, it is
94 interesting to investigate the relationship between environmental and gill-associated ammonia
95 oxidizers. Whenever environmental samples were available, the presence of AOB in the
96 surrounding water was checked (Table S2). In several studies, there was considerable overlap
97 between the environmental and gill AOB community. In particular, there seem to be few gill-
98 *specific* AOB in recirculating aquaculture systems (RAS) (Schmidt et al., 2016). This is of interest,
99 considering the reliance of RAS on active ammonia-oxidizing microorganisms in the biofilter
100 (Minich et al., 2020). However, in other datasets, gill-associated nitrifiers were identified while
101 none were found in the surrounding water (Elsheshtawy et al., 2021; Clokie et al., 2022), which
102 suggests selective colonization by gill-*specific* AOB. In all, we identified 115 AOB ASVs that were
103 present in gill samples but not in the environment (water, sediment or rearing system),
104 constituting the majority of ASVs found (Fig. 2, Table S2). While the lack of AOB observed in
105 environmental samples is not necessarily evidence of their complete absence, this observation
106 suggests that specific ammonia oxidizers are colonizing the gills.

107 While several studies have noticed gill-associated AOB in the past, most studies included
108 here do not mention their presence in the gill. A reason for this could be the low relative
109 abundance of gill-associated nitrifiers. Although the relative abundance reached over 50% in
110 some gill samples of Atlantic salmon (*Salmo salar* L.) (Birlanga et al., 2022), the relative
111 abundance in the investigated datasets is low (< 1%). This does not necessarily translate to a
112 small impact, as it was shown that nitrogen cycle microorganisms impact the nitrogen balance of

113 fish significantly (Mes et al., 2023a), regardless of the apparent low relative abundance (Mes et
114 al., 2023b). It is known that sampling methods can reduce the apparent abundance of
115 microorganisms in tissues with low bacterial biomass (Clokie et al., 2022). In order to assess the
116 true functional impact of gill-associated AOB on the fish, it will be key to relate AOB abundance
117 to measurable ammonia-oxidizing activity. Additional approaches are required to link the
118 abundance of nitrifying bacteria to their activity and establish the impact on fish physiology.

119 Considering the widespread presence of gill-associated AOB in teleost fish species, it is
120 important to establish the effects of these ammonia oxidizers on host health and nitrogen
121 physiology. Additionally, it is worthwhile to study the potential of the gill-associated AOB in
122 reducing ammonia accumulation, which is a common issue in aquaculture (Crab et al., 2007).
123 From a fundamental perspective, studying this functional interaction between nitrogen cycle 124 microorganisms and a vertebrate host is also highly interesting. These gill-associated AOB can
125 significantly affect fish nitrogen excretion and the widespread nature of this association suggests
126 that they can have similar impacts in more fish species. New studies should investigate the link
127 between abundance and activity of the nitrogen cycle bacteria in these fish species.

128

A

138
139
140

B

141
142
143
144
145
146
147
148
149
150
151
152

Fig. 1: Phylogenetic placement of gill-associated nitrifier amplicon sequencing variants (ASVs).
A: Phylogenetic placement of nitrifier ASVs (n = 49) from fish in marine habitats on a maximum likelihood full length 16S rRNA gene reference tree. ASVs clustering with a specific nitrifier lineage are collapsed into a cluster with the name of the nitrifier species. B: Phylogenetic placement of nitrifier ASVs (n = 133) from fish in freshwater habitats on a maximum likelihood full length 16S rRNA gene reference tree. ASVs clustering with a specific nitrifier lineage are collapsed into a cluster with the name of the nitrifier species. Phylogenetic placement was performed with *pplacer* and visualization was performed in iTOL.

153
154
155

156
157
158
159
160
161
162
163
164
165
166
167
168
169
170
171
172
173

Fig. 2: Phylogenetic placement of gill-specific nitrifier amplicon sequencing variants (ASVs). ASVs of gill-associated nitrifiers were compared to environmental nitrifiers to obtain ASVs that were only found in gill samples and not environmental samples. The remaining gill-specific ASVs (n=115) were placed on a full length 16S rRNA gene maximum likelihood tree. ASVs clustering in the *Nitrosomonas* genus are indicated in blue, while ASVs clustering in the *Nitrospira* genus are indicated in red. ASVs clustering with a specific nitrifier lineage are collapsed into a cluster with the name of the nitrifier species. Phylogenetic placement was performed with the pplacer algorithm and tree visualization was performed with iTOL.

174
175
176
177
178
179
180
181
182
183
184
185
186
187
188
189
190
191
192
193
194
195
196
197
198
199
200
201
202
203
204
205
206
207
208
209
210
211
212
213
214
215
216
217
218
219
220

References

- Birlanga, V.B., McCormack, G., Ijaz, U.Z., MacCarthy, E., Smith, C., and Collins, G. (2022). Dynamic gill and mucus microbiomes during a gill disease episode in farmed Atlantic salmon. *Scientific Reports* 12(1), 1-13.
- Callahan, B.J., McMurdie, P.J., Rosen, M.J., Han, A.W., Johnson, A.J.A., and Holmes, S.P. (2016). DADA2: high-resolution sample inference from Illumina amplicon data. *Nature methods* 13(7), 581.
- Clokie, B.G.J., Elsheshtawy, A., Albalat, A., Nylund, A., Beveridge, A., Payne, C.J., et al. (2022). Optimization of Low-Biomass Sample Collection and Quantitative PCR-Based Titration Impact 16S rRNA Microbiome Resolution. *Microbiology Spectrum* 10(6), e02255-02222.
- Crab, R., Avnimelech, Y., Defoirdt, T., Bossier, P., and Verstraete, W. (2007). Nitrogen removal techniques in aquaculture for a sustainable production. *Aquaculture* 270(1-4), 1-14.
- Elsheshtawy, A., Clokie, B.G.J., Albalat, A., Beveridge, A., Hamza, A., Ibrahim, A., et al. (2021). Characterization of external mucosal microbiomes of Nile tilapia and grey mullet co-cultured in semi-intensive pond systems. *Frontiers in Microbiology*, 3729.
- Ghanbari, M., Kneifel, W., and Domig, K.J. (2015). A new view of the fish gut microbiome: advances from next-generation sequencing. *Aquaculture* 448, 464-475.
- Legrand, T.P., Catalano, S.R., Wos-Oxley, M.L., Stephens, F., Landos, M., Bansemmer, M.S., et al. (2018). The inner workings of the outer surface: skin and gill microbiota as indicators of changing gut health in yellowtail kingfish. *Frontiers in microbiology* 8, 2664.
- Legrand, T.P., Wynne, J.W., Weyrich, L.S., and Oxley, A.P. (2019). A microbial sea of possibilities: current knowledge and prospects for an improved understanding of the fish microbiome. *Reviews in Aquaculture* 12(2), 1101-1134.
- Lorgen-Ritchie, M., Clarkson, M., Chalmers, L., Taylor, J.F., Migaud, H., and Martin, S.A. (2022). Temporal changes in skin and gill microbiomes of Atlantic salmon in a recirculating aquaculture system—Why do they matter? *Aquaculture*, 738352.
- Matsen, F.A., Kodner, R.B., and Armbrust, E. (2010). pplacer: linear time maximum-likelihood and Bayesian phylogenetic placement of sequences onto a fixed reference tree. *BMC bioinformatics* 11(1), 1-16.
- McFall-Ngai, M., Hadfield, M.G., Bosch, T.C., Carey, H.V., Domazet-Lošo, T., Douglas, A.E., et al. (2013). Animals in a bacterial world, a new imperative for the life sciences. *Proceedings of the National Academy of Sciences* 110(9), 3229-3236.
- Mes, W., Kersten, P., Maas, R.M., Eding, E.H., Jetten, M.S.M., Siepel, H., et al. (2023a). Effects of demand-feeding and dietary protein level on nitrogen metabolism and symbiont dinitrogen gas production of common carp (*Cyprinus carpio*, L.). *Frontiers in Physiology* 14.
- Mes, W., Lückner, S., Jetten, M.S., Siepel, H., Gorissen, M., and van Kessel, M.A. (2023b). Comparison of the gill and gut microbiomes of common carp (*Cyprinus carpio*) and zebrafish (*Danio rerio*) and their RAS environment. *Science of The Total Environment*, 165212.
- Minich, J.J., Poore, G.D., Jantawongsri, K., Johnston, C., Bowie, K., Bowman, J., et al. (2020). Microbial ecology of Atlantic salmon (*Salmo salar*) hatcheries: impacts of the built environment on fish mucosal microbiota. *Applied and Environmental Microbiology* 86(12), e00411-00420.

221 Norton, J.M. (2011). Diversity and environmental distribution of ammonia-oxidizing bacteria.
222 *Nitrification*, 39-55.

223 Perry, W.B., Lindsay, E., Payne, C.J., Brodie, C., and Kazlauskaitė, R. (2020). The role of the gut
224 microbiome in sustainable teleost aquaculture. *Proceedings of the Royal Society B*
225 *287*(1926), 20200184.

226 Pratte, Z.A., Besson, M., Hollman, R.D., and Stewart, F.J. (2018). The gills of reef fish support
227 a distinct microbiome influenced by host-specific factors. *Appl. Environ. Microbiol.*
228 *84*(9), e00063-00018.

229 Quast, C., Pruesse, E., Yilmaz, P., Gerken, J., Schweer, T., Yarza, P., et al. (2012). The SILVA
230 ribosomal RNA gene database project: improved data processing and web-based
231 tools. *Nucleic acids research* *41*(D1), D590-D596.

232 Schmidt, V., Amaral-Zettler, L., Davidson, J., Summerfelt, S., and Good, C. (2016). Influence of
233 fishmeal-free diets on microbial communities in Atlantic salmon (*Salmo salar*)
234 recirculation aquaculture systems. *Applied and Environmental Microbiology* *82*(15),
235 4470-4481.

236 Sehnaal, L., Brammer-Robbins, E., Wormington, A.M., Blaha, L., Bisesi, J., Larkin, I., et al.
237 (2021). Microbiome composition and function in aquatic vertebrates: small
238 organisms making big impacts on aquatic animal health. *Frontiers in microbiology* *12*,
239 358.

240 van Kessel, M.A.H.J., Mesman, R.J., Arshad, A., Metz, J.R., Spanings, F.A.T., van Dalen, S.C.M.,
241 et al. (2016). Branchial nitrogen cycle symbionts can remove ammonia in fish gills.
242 *Environmental microbiology reports* *8*(5), 590-594.
243
244

We would like to thank both reviewers for their constructive suggestions to improve our manuscript and have addressed their comments to our best capabilities. In line with both reviewers comments regarding missing metadata about the studies we used for our analysis, we have revised Supplementary table 1 to contain all the necessary information. We have also clarified and added several sentences in the main text to support our conclusions, while staying within the word limit of the format. We hope that these revisions have addressed the reviewer's comments. Below, the point-by-point responses to the comments of both reviewers are provided.

Reviewer #1 (Comments for the Author):

The authors use published datasets to identify the presence of Nitrosomonas in the fish gills in order to delineate whether symbiosis between those groups and fish gills is widespread among the teleost group. Overall I think it is a nice approach, but more information is needed before drawing conclusions. Line 65 and Supplementary Table S1: Which variable region did the authors choose for their analysis? Were all regions included? Please mention in the table which regions were targeted in the different studies and how this was taken into account during analysis. This is because the region can also have a certain bias in the results in terms of diversity and abundance.

We thank the reviewer for pointing out that the regions of the 16S rRNA genes included in the analysis were not clearly stated. We have added a column in table S1 in which the primers used and the region targeted are stated per study. We included all variable regions of the 16S rRNA gene targeted by studies and by aligning the short-length ASVs to a full-length reference dataset of Nitrosomonas 16S rRNA sequences, we were able to compare ASVs from different 16S rRNA regions based on closeness to reference strains. This approach has been used before in studies that combine results from amplicon sequencing of different 16S rRNA regions (Anderson et al., 2021). We clarified our approach in the main text and explicitly state that the approach allows us to compare different 16S rRNA regions (lines 74-76). We hope that this addresses the reviewer's comment.

Also which of all these studies refer to wild or farmed fish? This is not clear, and I do think it may have a significant effect on the outcome you present, as well as whether the fish come from RAS or flow trough systems, where the dilution effect is even larger in terms of finding out Nitrosomonas in the surrounding environment.

We agree that it is possible that the rearing environment can affect the presence of gill-associated Nitrosomonas, in particular when these microorganisms are present in biofilters. We have included a column in table S1 to indicate the source of the gill samples (laboratory, wild or aquaculture system, as well as the type of system when this information was provided in the original articles (flowthrough, RAS, cage, pond). We more closely examined the three studies in which RAS and flowthrough systems were directly compared (Minich et al., 2020; Minich et al., 2021; Quezada-Rodriguez et al., 2023) and tested whether there were notable differences in presence of AOB between the RAS and flowthrough systems. In both the fish and the surrounding systems, AOB could be found in similar numbers. Therefore, we find no clear pattern in the dataset that shows a difference between RAS and FT systems in terms of AOB presence. It would require additional studies that more directly examine the abundances and phylogeny of AOB in different compartments to test the reviewer's suggestion. Based on the total gill-associated AOB tree, we also cannot clearly see separation of taxonomy based on RAS or FT systems. We have added a sentence (lines 90-93) to address the lack of relationship of AOB presence with the type of aquaculture system. We hope this addresses the reviewers comment sufficiently.

I am also missing information in the tree on whether there is a phylogenetic closeness between biofilter/water Nitrosomonas of the study for example and the gills in RAS. I am not sure if you have such studies in your dataset.

We thank the reviewer for pointing out the interesting suggestion of comparing environmental AOB and those in the gills. In a previous manuscript (Mes et al., 2023), we did indeed see differences in AOB phylogeny between gill-associated and biofilter/water-associated AOB based on marker gene analysis. At the present, we did not include this in the analysis as not all studies included biofilter or water samples. We briefly address the overlap and/or lack of overlap between gill- and environmental AOB in several studies in the main text (lines 102-110). This focuses on those gill-associated AOB that are also in the environment, rather than all possible environmental AOB. While it is certainly an interesting approach to compare all environmental AOB, the article type limits the manuscript length and number of figures and we limited the scope of our manuscript to the overall pattern of gill-associated and gill-specific AOB presence. We hope that this addresses the point the reviewer made sufficiently.

Reviewer #2 (Comments for the Author):

Thank you for the opportunity to review this observation of very intriguing findings regarding the presence of ammonia oxidizers associated with teleost fish gills. Overall, the manuscript was very well-

written and the methodological approach appears to be valid, yielding findings of broad interest to the ASM community. I have made a few minor suggestions in the attached manuscript.
Change to "cycling" (nitrogen cycling symbionts)
Aquacultured instead of cultured
Change to "sequence"

We have adjusted our manuscript in the places where these words were used in line with the reviewer's suggestions.

There may be a way around this which could yield additional information. Other studies have been able to perform meta analyses in this manner and may be worth checking out.

We thank the reviewer for the suggestion to check meta-analyses of microbiome datasets and have found several meta-analyses of aquaculture microbiomes that show general comparisons of diversity indices (Cornejo-Granados et al., 2018; Cao et al., 2024) and the clear effect that certain factors (diet, rearing system, species) can have on microbiome diversity parameters, but we did not find meta-analyses focused on specific members of the microbiome and their presence/relative abundances. We included more metadata from the studies we used in our supplementary table 1 and see no particular factor that has large effects on the presence/number of Nitrosomonas ASVs. We do see the effect of water salinity on AOB phylogeny that we have discussed in the main text. We believe that once more gill microbiome studies become available, more statistically robust patterns can be observed regarding gill-associated AOB.

Has there been any work to determining the physical location of these potential epi- or endo- symbionts in relation to host gill tissues?

We have previously showed the location of ammonia-oxidizing and denitrifying symbionts via fluorescent in situ hybridization in common carp gills (van Kessel et al., 2016). In the case of carp, the symbionts were endosymbiotic and located close to large blood vessels in the gill tissue, and this is a highly interesting feature of the symbiosis. Unfortunately, the studies analyzed here did not include microscopical analysis of bacterial symbionts and thus we can only speculate on whether the location of the gill-associated AOB we identify in the current datasets is similar. Follow-up research should confirm this location for other species in order to establish the location. Since it is currently unknown and we are limited in words, we believe the possible location of symbionts is outside the scope of this Observation article.

References

- Anderson, C.J., Koester, L.R., and Schmitz-Esser, S. (2021). Rumen epithelial communities share a core bacterial microbiota: a meta-analysis of 16S rRNA gene Illumina MiSeq sequencing datasets. *Frontiers in Microbiology* 12, 625400.
- Cao, S., Dicksved, J., Lundh, T., Vidakovic, A., Norouzitalab, P., and Huyben, D. (2024). A meta-analysis revealing the technical, environmental, and host-associated factors that shape the gut microbiota of Atlantic salmon and rainbow trout. *Reviews in Aquaculture*.
- Cornejo-Granados, F., Gallardo-Becerra, L., Leonardo-Reza, M., Ochoa-Romo, J.P., and Ochoa-Leyva, A. (2018). A meta-analysis reveals the environmental and host factors shaping the structure and function of the shrimp microbiota. *PeerJ* 6, e5382.
- Mes, W., Lückner, S., Jetten, M.S., Siepel, H., Gorissen, M., and van Kessel, M.A. (2023). Comparison of the gill and gut microbiomes of common carp (*Cyprinus carpio*) and zebrafish (*Danio rerio*) and their RAS environment. *Science of The Total Environment*, 165212.
- Minich, J.J., Nowak, B., Elizur, A., Knight, R., Fielder, S., and Allen, E.E. (2021). Impacts of the marine hatchery built environment, water and feed on mucosal microbiome colonization across ontogeny in Yellowtail Kingfish, *Seriola lalandi*. *Frontiers in Marine Science* 8, 516.
- Minich, J.J., Poore, G.D., Jantawongsri, K., Johnston, C., Bowie, K., Bowman, J., et al. (2020). Microbial ecology of Atlantic salmon (*Salmo salar*) hatcheries: impacts of the built environment on fish mucosal microbiota. *Applied and Environmental Microbiology* 86(12), e00411-00420.
- Quezada-Rodriguez, P.R., Taylor, R.S., Jantawongsri, K., Nowak, B.F., and Wynne, J.W. (2023). Association between melanin deposits in gill tissue and microbiome across different hatchery reared Atlantic salmon. *Journal of Applied Microbiology*.
- van Kessel, M.A.H.J., Mesman, R.J., Arshad, A., Metz, J.R., Spanings, F.A.T., van Dalen, S.C.M., et al. (2016). Branchial nitrogen cycle symbionts can remove ammonia in fish gills. *Environmental microbiology reports* 8(5), 590-594.

Re: Spectrum00295-24R1 (Gill-associated ammonia oxidizers are widespread in teleost fish)

Dear Dr. Wouter Mes:

Your manuscript has been accepted, and I am forwarding it to the ASM production staff for publication. Your paper will first be checked to make sure all elements meet the technical requirements. ASM staff will contact you if anything needs to be revised before copyediting and production can begin. Otherwise, you will be notified when your proofs are ready to be viewed.

Sincerely,
Luke Iwanowicz
Editor
Microbiology Spectrum